# Intensive-Use-Oriented Performance Evaluation and Optimization of Rural Industrial Land: A Case Study of Wujiang District, China

Xiaojun Ye [1], Lingyun Fan [2] and Cheng Lei [1,3,*]

1 Gold Mantis School of Architecture, Soochow University, Suzhou 215123, China; 20224041001@stu.suda.edu.cn
2 School of Architecture and Urban Planning, Suzhou University of Science and Technology, Suzhou 215129, China; fly@usts.edu.cn
3 Center for Chinese Urbanization Studies, Soochow University, Suzhou 215008, China
* Correspondence: leicheng@suda.edu.cn; Tel.: +86-15312186156

**Abstract:** Rural industrialization is one of the core drivers of urban and rural spatial evolution and economic transformation in China. Given the background of stock and reduction planning, the development of rural industrial land, which has long relied on land inputs to increase production and inefficient expansion, is facing severe constraints and challenges. How to improve the spatial performance of rural industrial land and promote industrial upgrading and intensive land use have become vital issues for the healthy development of rural areas. This paper draws upon smart shrinkage theory to provide an analytical framework for the intensive-use-oriented performance evaluation of rural industrial land, unlike the evaluation method of efficiency orientation for industrial land, which emphasizes the core goal of the input and output of production factors per unit area. Based on the analysis framework, this study explored the parcel-microscale performance evaluation methods for rural industrial land, and the evaluation index system construction covers the four dimensions of economic performance, social performance, ecological performance, and land use structure performance. Wujiang District of Suzhou City was used as a case study to carry out a comprehensive performance evaluation and analyze the differences in RILP in space and industry. Based on the evaluation results, the key problems of rural industrial land were identified, and corresponding optimization strategies for rural industrial land are proposed from the aspects of land use control, spatial agglomeration, and industrial upgrading. This study intended to address the current major national strategic needs and solve the real dilemmas faced in the process of rural industrial land development. It is hoped that the study will provide a theoretical reference for the transformation of rural industrial land and policy-making for rural revitalization.

**Keywords:** rural industrial land; intensive use; smart shrinkage; performance evaluation; parcel-microscale; optimization

## 1. Introduction

Industrialization is the primary driver of urbanization, and the process of rapid industrialization has accelerated the development of cities, which is crucial for economic growth and urban functional structure optimization [1,2]. With the continuous acceleration of China's industrialization and urbanization processes, the rapid expansion of construction land, especially industrial land, has resulted in the occupation of large amounts of arable land and forestland resources; this poses a serious threat to the security of grain and the safety of the ecological environment [3–5]. The existing literature has revealed that the scale of construction land in some large cities has approached the limit, the expansion of industrial land coexists with inefficient use [6], and the development model mainly characterized by land consumption needs to be changed urgently [7].

With the continuous promotion of territorial planning reform, China's urban construction model is gradually turning from "incremental planning" to "stock and reduction planning", and "strictly controlling incremental quantity and revitalizing stock" has become the core means by which to achieve high-quality development. Intensive industrial land use arises at the historic moment under this background [1]. Intensive industrial land use is an effective method by which to promote intensive industrial land use and strengthen land macro-control, and it is also an important indicator used to measure the spread and spatial compactness of industrial land [8]. Thus, in the process of rapid urbanization, how to effectively improve the utilization efficiency and output effectiveness of industrial land within a limited geographical space has become a key issue in promoting the spatial transformation of China's urban and rural areas and the sustainable development of the economy.

Studies on industrial land performance evaluation date back to the study of Land-tax Theory in Classical Economics, which is rich in content, comprising various scales, multidimensional systems, and multiple methods. Most existing studies focus on the performance evaluation of different utilization methods, such as the evaluation of land use economic efficiency and construction land sustainable development [9–12], and on this basis, they have been extended to industrial land use efficiency evaluation [13–20] and intensive industrial land use evaluation [21–23]. The research scale mainly focuses on the macro and middle evaluation of various spatial ranges such as provinces, urban clusters, cities, and parks [13–23]. Through analyzing the performance levels of different scales and their differences [13,14,24], we can reveal the influence mechanisms of industrial land performance among regions [14–16,18,20] and try to explore the formulation of management policies [25]. However, there are few discussions on the more refined micro-scale, which uses land parcels as evaluation units. The evaluation system dimensions place more emphasis on input and output efficiency, as well as the economic, social, and ecological benefits of industrial land [26,27], and gradually establish a multidimensional performance evaluation index system. Existing research methods mainly adopt the analytic hierarchy process, the eighteen equal divisions method, the composite index method, the Delphi method, and the entropy method [6,28]; apply the Bayesian discriminant function method [29]; verify the responsibility location method by compiling performance codes [30]; and use stochastic frontier analysis [13], data envelopment analysis [31], artificial neural networks [32], etc.

Since China's reform and opening up, "rural industrialization" has become the epitome of its early- and mid-term industrialization and urbanization, constituting the internal driving force and spatial development logic for the rapid development of urban and rural areas. Mainly motivated by township and village enterprises (TVEs), the "bottom-up" process of rural industrialization has led to a low threshold for industrial land access [33], which has resulted in rapid development of the rural non-agricultural industry and local transfer of the rural labor force [34]. The reform of the national economic system in the 1990s resulted in a large number of foreign enterprises settling in, and TVEs were forced to reform. The official preference for foreign-owned enterprises led to many TVEs becoming bankrupt in the following economic crisis, or being restructured into private enterprises [35], and these became an important part of the Pearl River Delta and Yangtze River Delta private economy. According to statistics, rural industrial gross output accounted for about half of the total output of the national economy in 2013 [36]. However, urbanization and industrialization over the past several decades have led to the gradual replacement of the traditional Chinese countryside with residential blocks and scattered factory buildings [33]. The industrial economy and industrial spatial dominance have resulted in a large amount of rural industrial land, which profoundly affects the development path of rural areas. The long-term reliance on land input to increase production and the inefficient spread of the crude growth of the rural industry have also led to a series of problems, such as spatial fragmentation, social differentiation, and uncontrolled planning, revealing the deep-rooted root cause of the "structural inefficiency" of rural industrial land. As rural industrial land is the primary location for economic activities of the rural population and industrial development in rural areas [37,38], the issue of its intensive and efficient use has received widespread attention.

Responding to the acute contradiction between land expansion and inefficient land use due to rapid rural industrialization and in situ urbanization, initiatives such as smart shrinkage [39–41], compact development [42], and other sustainable development ideas and planning methods have been implemented. Smart shrinkage theory aims to ensure the rational withdrawal and optimal reorganization of resources, and the respective reduction of the amount and scale of land, as well as improving intensive land use, promoting industrial transformation and upgrading, and ensuring the healthy and sustainable development of urban and rural environments [41]. Industrial transformation and upgrading, intensive land use, and cultivated land protection policies have brought about the tightening of urban and rural construction land quotas, which have jointly driven the transformation of rural industrial land use [34,43]. Therefore, against the background of stock and reduction planning, smart shrinkage theory provides a new perspective for this study, and it is important to establish a new evaluation and optimization method oriented toward the intensive use of rural industrial land.

In summary, performance evaluations are relatively short in micro-scale studies at the rural level, and there is a lack of comprehensive and accurate evaluation for rural industrial land transformation. In addition, existing studies on rural industrial land mainly focus on land distribution, expansion characteristics, and formation mechanisms [37,38,44–47], and mostly study the layout of rural industrial land at the macro level, with less research on performance measurement and evaluation at the micro level. In view of this, based on the analysis framework of smart shrinkage theory, in this study, we establish a multidimensional comprehensive performance evaluation system for rural industrial land. We take Wujiang District as an empirical case by which to analyze the reasonable distribution of RILP in terms of industry, land, and space, and put forward targeted strategies to optimize rural industrial land. It is hoped that the study will provide a theoretical reference for local governments in formulating criteria for determining inefficient industrial land and making policies for transforming differentiated rural industrial land.

The rest of this article is organized as follows. Section 2 provides the analytical framework for evaluating rural industrial land in terms of its performance based on smart shrinkage theory. Section 3 outlines the research materials and methodology. Section 4 details the results of the case study on rural industrial land. Section 5 offers the policy implications and optimization strategies for rural industrial land, and Section 6 outlines the research conclusions.

## 2. Analysis Framework for the Performance Evaluation of Rural Industrial Land Based on Smart Shrinkage Theory

The term "smart decline" originated from Germany's management and operation model for relatively poor and dilapidated cities, aiming to solve the economic and physical environmental problems in urban operation caused by population decline [48]. The concept of smart shrinkage was first introduced by Professor Frank Popper of Rutgers University in 2002 and summarized as "Planning for less—Fewer people, fewer buildings, fewer land uses" [39]. Since then, smart shrinkage theory as a planning strategy has been widely disseminated and promoted for urban planning in the US and worldwide. Examples include the Queen City Master Plan developed by the City of Farrow in 2006, Detroit's nonprofit campaign to control property consolidation in 2008, and Ohio's Citywide Plan for Youngstown in 2010.

With the rapid development of China's urbanization process, rural shrinkage is a current development reality and an inevitable trend [40,41]. On the one hand, the continuous migration of rural populations to cities and towns leads to the hollowing out and desertion of rural areas [49], resulting in "passive shrinkage". On the other hand, driven by the two wheels of new urbanization and rural revitalization, national and local governments at all levels actively advocate the intensive and efficient use of land and promote the "active shrinkage" of rural areas. However, the current rural shrinkage is not sensible or normal [50]. According to statistics, China's rural permanent population is decreasing at an

annual rate of 1.6%, while the urban environment is increasing at an annual rate of 1% [51]. Therefore, domestic scholars have introduced smart shrinkage theory, which is very similar to the current situation of "human shrinkage and land expansion" in rural China, into the study of rural human residential space [40], and put forward smart shrinkage [41], reduction planning [50], and other countermeasures. Rural smart shrinkage is the process of improving rural vitality by means of functional optimization and spatial agglomeration [41]. In China, following the decrease in the rural population and labor force and the corresponding change in the way rural production is organized [40,41], there is an urgent need to rationalize the withdrawal and optimize the reorganization of rural habitat resources, to optimize the spatial structure of the countryside, and to promote high-quality urban and rural development. Essentially, rural smart shrinkage is expected to solve socioeconomic transformation problems, such as inefficient land sprawl caused by excessive urban and rural construction land, improve land-use efficiency, improve the habitat environment, enhance accessibility, protect the ecological environment, reduce energy consumption, and strengthen urban renewal [40,41,52]. This will facilitate the compact growth of rural spaces, effective protection of cultivated land resources and the ecological environment, and sustainability of urban and rural habitats, and thus represents a compact, intensive, and efficient rural shrinkage approach. Smart shrinkage theory therefore provides a new perspective for the intensive-use performance evaluation and layout optimization of rural industrial land.

This study draws on the concepts, principles, and approaches of smart shrinkage theory, holding that the realization strategies of rural industrial land intensive use include four aspects. First, they include promoting the centralized distribution of rural industrial land to improve industrial land's economic and output efficiency under the scale and agglomeration economies [1]. Second, they improve the habitat environment and accessibility of rural industrial land. Convenient and efficient transportation and a high-quality habitat environment are often more likely to attract the diverse talent needed for the development of rural industrial land [53]. Third, they protect the ecological environment and reduce energy consumption. Through industrial transformation upgrading, eliminating low-level production technology, and reducing the environmental damage caused by rural industrial development to surrounding areas, we can realize the coordination of industrial development and ecological environmental protection [54]. Fourth, they realize intensive and efficient industrial space through compact development, and the three-dimensional development of rural industrial land is emphasized through indices such as the floor area ratio, building density, and land area. In addition, various issues, such as ownership and the building life of rural industrial land, are considered.

In summary, the analytical framework of the "performance evaluation" of rural industrial land can be built based on smart shrinkage theory to study rural industrial land intensive use against the background of stock and reduction planning. Improving economic and output efficiency, strengthening transportation accessibility, enhancing industrial upgrading, and promoting the centralized distribution of rural industrial land through the performance evaluation of rural industrial land intensive use promote rural industrial land's economic, social, ecological, and land use structure benefits. Specifically, the dimensions of intensive-use-oriented rural industrial land performance evaluation are as follows: economic performance, social performance, ecological performance, and land-use structure performance. In this study, based on smart shrinkage theory, we propose an analytical framework for rural industrial land performance evaluation (Figure 1) and then build an evaluation index system based on this framework.

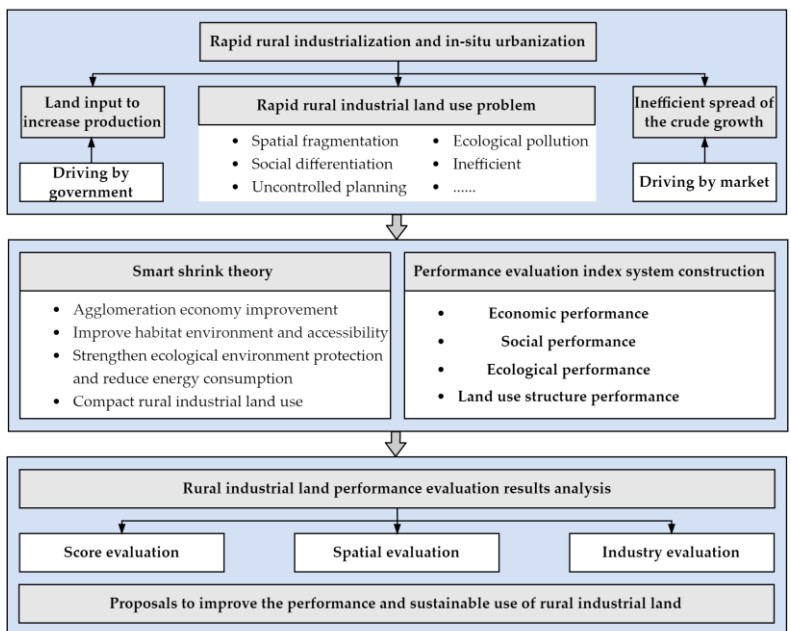

**Figure 1.** The research framework for rural industrial land performance evaluation.

## 3. Materials and Methods

### 3.1. Study Area

Wujiang District of Suzhou City is located in the southeast of Jiangsu Province and the middle of the Yangtze River Delta (30°45′36″ N~31°13′41″ N, 120°21′04″ E~120°53′59″ E), bordering Shanghai to the east, Taihu Lake to the west, Jiaxing and Huzhou to the south, and Suzhou's main urban area to the north, covering an area of 1237.48 km² (Figure 2). By the end of 2020, the resident population had risen to 1,540,000, with 876,700 people registered, resulting in a yearly regional GDP of 2002.83 billion yuan and a per capita GDP of 129,600 yuan. The territory is governed by four streets, seven townships, 235 administrative villages, 5560 village groups (social teams), and a rural population of 383,900 individuals.

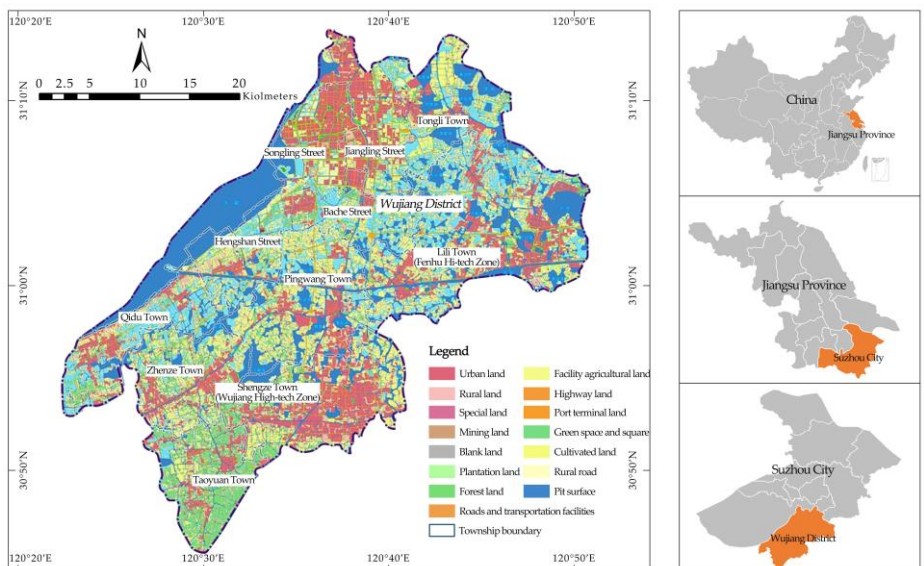

**Figure 2.** Location and current land use status of Wujiang.

The study area is an example of the rapid economic development of China's first rural industrialization, leading to in situ urbanization. In the 1980s, with the transfer of manufacturing from Shanghai, rural industry flourished and quickly became an important economic pillar

of the region. With the inclusion of Wujiang District into the Yangtze River Delta Ecological Green Integrated Development Demonstration Zone, the concept of "green economy and sustainable development" has resulted in higher requirements for the industrial development of the area; Wujiang District is therefore suitable as a case study area.

### 3.2. Data Source and Processing

The data involved in the study are mainly from the 2020 Wujiang District Industrial Land Survey Database, which includes information on land location, land area, enterprise output value, tax revenue, building density, and floor area ratio. The current data situation is good and can reflect the interrelationship between ground objects to the maximum extent [55]. The study data also include the boundaries of urban areas, townships, and development zones, derived from the territorial and spatial planning of Wujiang District and the general planning of each township. Based on the road traffic planning of the whole district, the road types and grades were determined regarding Baidu Maps and formed after topological inspection by ArcGIS 10.4 software.

After deducting the industrial land located in the central urban area, various townships, and development zones of Wujiang District, we obtained the basic unit of performance evaluation (i.e., each rural industrial enterprise land plot) based on the characteristic that all industries located in rural areas can be classified as rural areas [45]. Based on this, the 2020 remote sensing image and Baidu Map data were used to calibrate the information and check the gaps in the database of rural industrial land according to the characteristics of industrial enterprises (such as floor area, roof color, and contour shape) (Figure 3). The final result was 4780 rural industrial land plots in the region, with a total area of 45.87 km$^2$ (Figure 4).

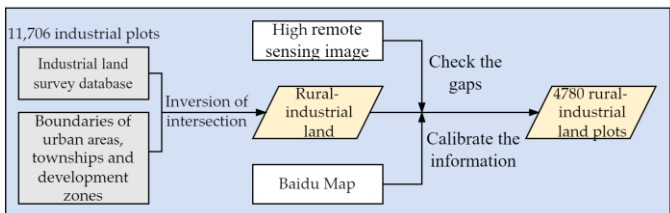

**Figure 3.** Rural industrial land data acquisition process.

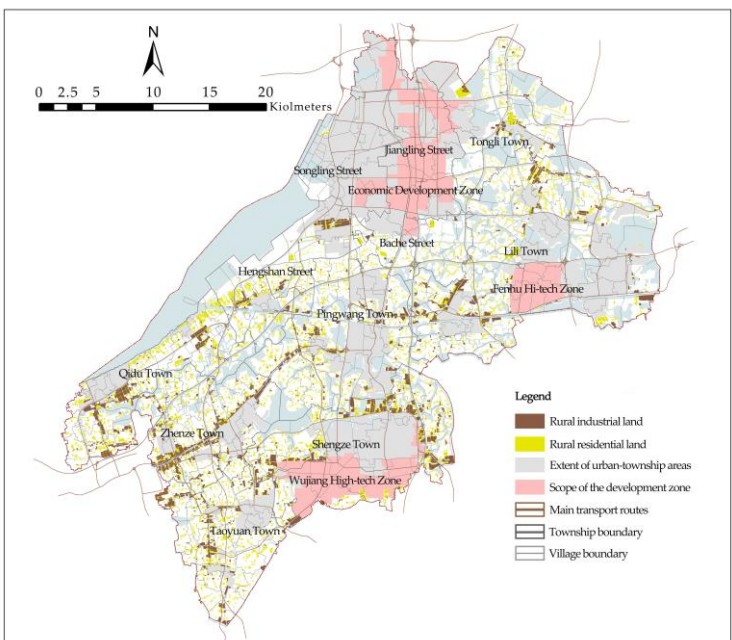

**Figure 4.** Current distribution of rural industrial land in the study area.

In 2020, the total output of rural industrial land in Wujiang District was 87,055 billion yuan, creating 21.5% of industrial output with 32% of the district's land area. The current building density of rural industrial land in the study area is 51.88%, and the building volume ratio is 0.70. The aforementioned parameters basically meet the requirements of the Jiangsu Province Construction Land Index (2014 edition) (building density of >40% and building volume ratio of >0.7). The intensive use of rural industrial land in the study area is low, and there is considerable potential for development. Significant spatial distribution pattern differences also exist regarding rural industrial land use in the study area. Specifically, Shengze Town has the highest building density (55.61%), whereas Tongli Town has the lowest (35.62%), indicating that the rural industrial land in this area is largely unused and wasted. Lili Town has the highest floor area ratio (0.79), whereas Taoyuan Town has the lowest floor area ratio (0.49), mainly because most of the rural industrial enterprises in this area are old factories with scattered spatial layouts.

### 3.3. Evaluation Index System Construction and Method

The purpose of performance evaluation is to grasp the efficiency of economic activities and provide a basis for the government to formulate land use policies and industrial development policies [30]. The rural industrial land performance (RILP) is a comprehensive evaluation index used to quantitatively measure the status of the development level and quality of rural industrial land. The value of the RILP quantitatively describes the multidimensional performance level of rural industrial land in the process of socioeconomic development. Rural industrial land transformation is essential to ensure better adjustment of the land use transition process and trend, as well as the promotion of the social economy and ecological environment in rural areas [38,56]. From this aspect, the RILP can be seen not only as the result of multiple factors, such as economy, society, and ecology. Therefore, it is necessary to consider multidimensional evaluation indices and carry out targeted index selection and quantitative classification [57,58].

Based on the literature, expert consultation, available data sources, and harmonizing the principles of scientificity, comprehensiveness, typicality, and operability, we selected 17 indices from the 4 dimensions to construct a parcel-microscale performance evaluation index system. First, economic performance is an essential attribute by which to measure the development level and quality of rural industrial land. We focused on the gold content of land use and selected four indices (C1–C4), including output capacity and output efficiency. Second, previous studies have shown that in the stage of socioeconomic transformation and development, people's demands for convenient and efficient transportation and a high-quality living environment are growing [53]. We chose C5–C8 to measure the social performance of rural industrial land by using measurement methods such as kernel density estimation, buffer analysis, and network analysis. Third, low ecological consumption is the basic requirement for ecological civilization. Waste gas, waste residue, wastewater, and other undesired outputs in the process of industrial production will lead to environmental pollution [59], and these industrial enterprises are controlled in accordance with regulations. We chose C9–C12 to represent the ecological performance. Fourth, rural industrial land intensive use emphasizes the compact use and spatial three-dimensional development of rural industrial land. Based on this scheme, we selected five indices (C13–C17) to measure the level of land use structure performance of rural industrial land (Table 1). In this study, the appropriateness of variable selection was measured based on the principles of KMO > 0.7 and Bartlett's spherical test ($p < 0.01$). The results show that KMO = 0.93 and $p < 0.001$, and the selected indices were suitable for generalized principal component analysis.

**Table 1.** The performance evaluation index system of rural industrial land.

| Target Layer | First-Level Indices | Second-Level Indices | Third-Level Indices | Calculation or Description of the Indices | Index Code | Analytic Hierarchy Process | Entropy Method | Combined Weight |
|---|---|---|---|---|---|---|---|---|
| Rural industrial land performance (RILP) | Economic performance | Economically efficient | Total output value | Annual output value of industrial land | C1 | 0.074 | 0.074 | 0.074 |
| | | | Total profit and tax | Annual profit tax on industrial land | C2 | 0.075 | 0.069 | 0.071 |
| | | Efficient output | Average industrial output value | Annual output value of industrial land/land area | C3 | 0.090 | 0.077 | 0.084 |
| | | | Average industrial profit tax | Annual profit tax on industrial land/land area | C4 | 0.090 | 0.067 | 0.078 |
| | Social performance | Suitable for industry and development | Land per employee | Number of employees in industrial enterprises/land area | C5 | 0.067 | 0.075 | 0.071 |
| | | | Accessibility | Distance from land i to the edge of the nearest traffic artery | C6 | 0.055 | 0.029 | 0.042 |
| | | Spatial layout | Location suitability for development | Evaluation assignment | C7 | 0.032 | 0.192 | 0.112 |
| | | | Mean nearest distance (C8) | Average of the distances from land i to its nearest industrial land | C8 | 0.055 | 0.034 | 0.045 |
| | Ecological performance | Ecological control | Ecological Protection Red Line | Evaluation assignment | C9 | 0.039 | 0.019 | 0.029 |
| | | Pollution energy consumption | Emissions of pollutants per unit of industrial output value | Annual pollutant emissions/annual output value of industrial land | C10 | 0.062 | 0.060 | 0.061 |
| | | | Consumption of gas per unit of industrial output value | Annual gas consumption/annual output value of industrial land | C11 | 0.048 | 0.052 | 0.050 |
| | | | Electricity consumption per unit of industrial output value | Annual electricity consumption/annual output value of industrial land | C12 | 0.054 | 0.048 | 0.051 |
| | Land use structure performance | Development intensity | Floor area ratio | Total industrial building area/land area | C13 | 0.066 | 0.036 | 0.051 |
| | | | Building density | Industrial building area/land area | C14 | 0.064 | 0.028 | 0.046 |
| | | | Land area | Industrial land area | C15 | 0.050 | 0.046 | 0.048 |
| | | Term of ownership | Nature of Property rights | Evaluation assignment | C16 | 0.035 | 0.073 | 0.054 |
| | | | Year of completion | Year of completion of industrial buildings, reflecting their old and new condition | C17 | 0.045 | 0.021 | 0.033 |

The weight of each evaluation indicator was determined using the analytic hierarchy process (AHP) and the entropy method. First, an analytic hierarchy model was constructed using Yaahp software, and a "two-two" comparison matrix was established. Twenty-three experts from universities, planning agencies, and government departments were invited to score the relative importance of the selected indicators in each dimension on a scale of 1 to 9, and the scores were normalized. The analysis results passed the consistency test. The z-score standardization method was chosen to eliminate differences in the scale, order of magnitude, and magnitude of change in the quantity of each evaluation indicator [60].

Next, the entropy method was used to determine the entropy value and objective weight of the $j$th evaluation index of the $i$th evaluation object. The calculation formula is as follows:

$$P_{ij} = X_{ij} / \sum_{i=1}^{m} X_{ij} (i = 1, 2, \ldots, m;\ j = 1, 2, \ldots, n) \tag{1}$$

$$E_j = -ln(n)^{-1} \sum_{i=1}^{m} P_{ij}\ ln(P_{ij}) (i = 1, 2, \ldots, m;\ j = 1, 2, \ldots, n) \tag{2}$$

$$W_j = (1 - E_j)j / \sum_{j=1}^{n} (1 - E_j) (j = 1, 2, \ldots, n) \tag{3}$$

where $P_{ij}$ is the $i$th ($i$ = 1, 2, $\ldots$, $m$) object in the $j$th ($j$ = 1, 2, $\ldots$, $n$) indicator, $X_{ij}$ is the quantity value of the $j$th indicator of the $i$th object, and $E_j$ and $W_j$ are the entropy value and weight of the $j$th indicator, respectively.

Finally, the basic weights calculated using the analytic hierarchy process and the entropy method were each weighted by 50% to obtain the total weight value of each indicator, and the performance evaluation results for rural industrial land were obtained after the linear weighting summation of each indicator. The calculation formula is as follows:

$$RILP = \sum_{i=1}^{m} W_i C_i \tag{4}$$

where $RILP$ is the rural industrial land performance value, $W_i$ is the weight value of each evaluation indicator, and $C_i$ is the standard value of each indicator, specifically, $C1 \sim C17$. The higher the RILP value (between 1 and 100), the higher the RILP.

## 4. Results

We analyzed the performance evaluation results of rural industrial land on a parcel-microscale from three aspects: score, spatial, and industry distribution. Score evaluation uses Jenks to classify the results into different performance grades. Spatial evaluation adopts global and local autocorrelation methods to analyze the spatial agglomeration degree of RILP. Industry evaluation summarizes and compares the average performance evaluation values of 30 manufacturing industries on rural industrial land.

### 4.1. Grade Classification of RILP

A single indicator can reflect a single physical characteristic of industrial land, and rural industrial land has multiple characteristics [57,58]. The performance evaluation needs to be based on a comprehensive analysis of multiple factors to determine the direction and timing of renewal. A total of 4780 rural industrial land performances were measured in Wujiang District. Figure 5 indicates that the RILP score is 100 out of 100; the lighter the color of the spot, the lower the performance score, and this suggests that the land should be renewed as soon as possible. Conversely, the darker the color of the spot, the higher the performance score; the industrial land attribute should be retained, and appropriate safeguards should be taken.

The performance evaluation scores were assigned to five low-to-high grades according to Jenks: low, medium-low, medium, medium-high, and high performance (Table 2). (1) The average land area of all grades significantly changed from low to high, gradually increasing from 0.78 hm$^2$ of low performance to 1.48 hm$^2$ of high performance, indicating that the grade division generally reflected the relationship between the performance level and the degree of industrial land scattered. With the improvement in land use performance, the corresponding rural industrial land sporadic degree decreases. (2) The grades covered areas of 1198.82, 828.61, 636.13, 999.28, and 924.19 hm$^2$ and accounted for 26.13%, 18.06%, 13.87%, 21.78%, and 20.15% of the total rural industrial land area in the study area, respectively. The findings thus indicate that there is considerable potential for rural industrial land to be

redeveloped and show that its performance evaluation is of great significance for boosting intensive use. In addition, the performance grades of rural industrial land with "sporadic distribution and small land area" are lower, while those of land with "agglomeration and cluster development" are higher. (3) From the distribution of the number of plots and the proportion of the area occupied by the corresponding grades, the number and proportion of medium performance are smaller than the low-medium and low performance of low performance and the high-medium performance of high performance. The number of plots involved in the performance grade and the area occupied show a dumbbell structure, which is close to the normal distribution characteristics, reflecting the relative rationality of the result of the grade division.

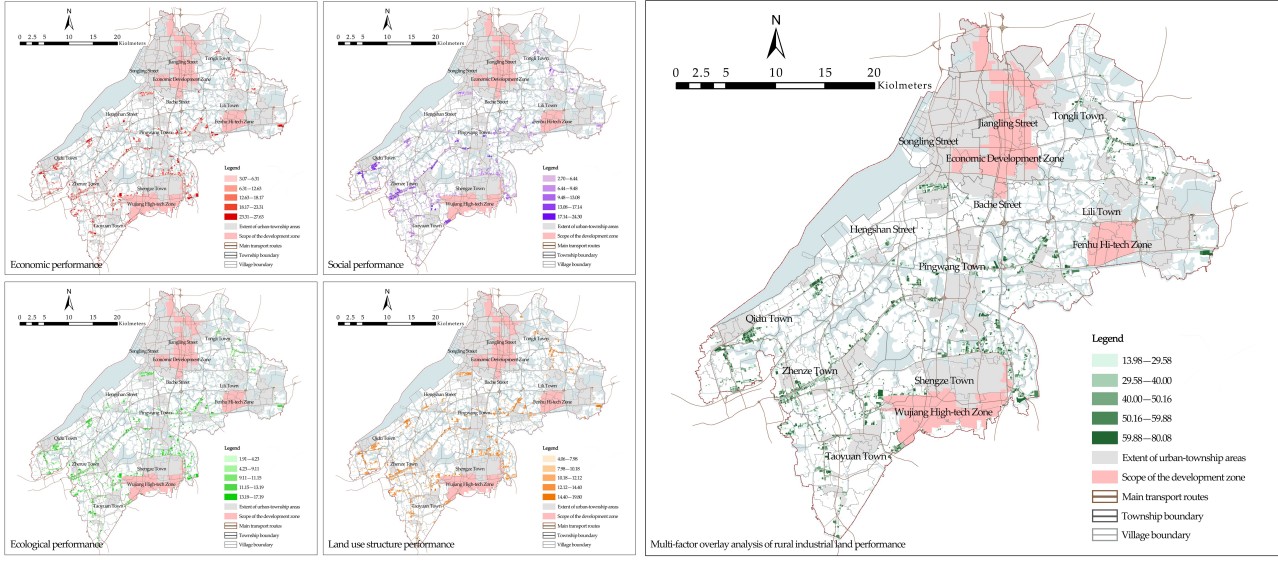

**Figure 5.** Multifactor analysis of RILP.

**Table 2.** Parcel information of RILP evaluation grading (hm$^2$) (number of blocks).

| Performance Grade | Range of RILP Values | Number of Parcel | Quantity Ratio | Average Parcel Size | Total Area | Area Ratio |
|---|---|---|---|---|---|---|
| Low performance | 13.98~29.58 | 1535 | 32.11% | 0.78 | 1198.82 | 26.13% |
| Medium-low performance | 29.58~40.00 | 907 | 18.97% | 0.91 | 828.61 | 18.06% |
| Medium performance | 40.00~50.16 | 756 | 15.82% | 0.84 | 636.13 | 13.87% |
| Medium-high performance | 50.16~59.88 | 957 | 20.02% | 1.04 | 999.28 | 21.78% |
| High performance | 59.88~80.08 | 625 | 13.08% | 1.48 | 924.19 | 20.15% |
| Total | 13.98~80.08 | 4780 | 100.00% | 5.05 | 4587.03 | 100.00% |

Table 3 indicates clear differences in the spatial distributions of the RILP grades. Low-performance rural industrial land is concentrated in Zhenze Town and Lili Town. The aforementioned areas contain 401 and 261 blocks, respectively, covering areas of 228.36 and 202.92 hm$^2$. With medium-low performance, Zhenze Town contains the highest number of these blocks (227) and covers an area of 177.12 hm$^2$. Medium-performance rural industrial land is mainly concentrated in Lili Town (178 blocks), Qidu Town (134 blocks), and Zhenze Town (123 blocks), covering areas of 112.90, 98.97, and 105.80 hm$^2$, respectively. With medium-high performance, Shengze Town contains the highest number of these blocks (211), followed by Lili Town (182 blocks). The 211 blocks in Shengze Town cover an area of 266.63 hm$^2$, and the 182 blocks in Lili Town cover an area of 176.88 hm$^2$. High-performance rural industrial land is mainly concentrated in Qidu Town (143 blocks) and Zhenze Town (104 blocks), covering areas of 181.14 and 122.60 hm$^2$, respectively. The spatial distribution of the RILP grading is displayed in Figure 6.

**Table 3.** Area (hm$^2$) of rural industrial land performance evaluation (number of blocks).

| Study Area | Low Performance | Medium-Low Performance | Medium Performance | Medium-High Performance | High Performance |
|---|---|---|---|---|---|
| Songling Street | 0 (0) | 0 (0) | 0 (0) | 0 (0) | 0 (0) |
| Hengshan Street | 137.11 (164) | 43.07 (56) | 46.78 (54) | 71.87 (65) | 28.64 (29) |
| Bache Street | 9.85 (11) | 3.40 (4) | 12.69 (4) | 3.98 (4) | 1.05 (2) |
| Jiangling Street | 0 (0) | 0 (0) | 0 (0) | 0 (0) | 0 (0) |
| Tongli Town | 35.65 (39) | 39.14 (35) | 8.36 (23) | 31.41 (29) | 16.47 (12) |
| Lili Town | 202.92 (261) | 133.17 (171) | 122.90 (178) | 176.88 (182) | 163.73 (93) |
| Pingwang Town | 82.49 (90) | 70.59 (60) | 53.58 (56) | 101.08 (85) | 138.69 (88) |
| Shengze Town | 174.75 (204) | 148.32 (131) | 100.30 (118) | 266.63 (211) | 175.44 (95) |
| Zhenze Town | 228.36 (401) | 177.12 (227) | 98.97 (134) | 127.16 (165) | 122.60 (104) |
| Qidu Town | 134.02 (193) | 102.05 (138) | 105.80 (123) | 93.84 (130) | 181.14 (143) |
| Taoyuan Town | 193.67 (172) | 111.76 (85) | 86.75 (66) | 126.43 (86) | 96.43 (59) |
| Total | 1198.82 (1535) | 828.61 (907) | 636.13 (756) | 999.28 (957) | 924.19 (625) |

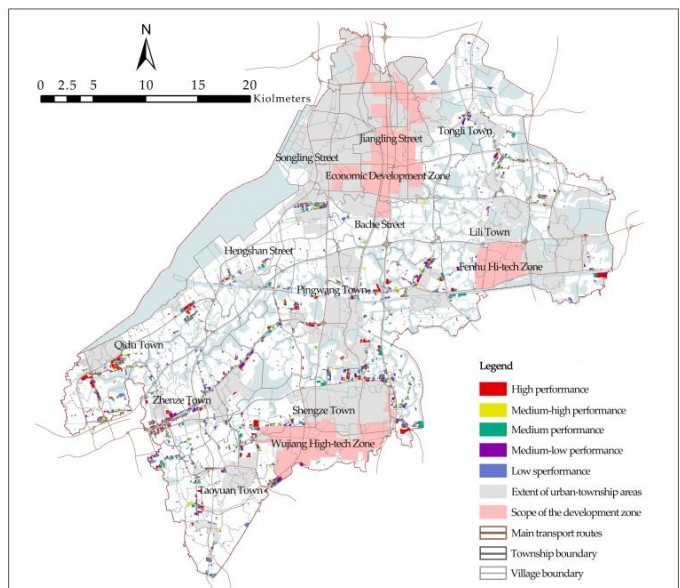

**Figure 6.** Spatial distribution of RILP grading.

### 4.2. Spatial Differentiation of RILP

Taking 4780 rural industrial plots in Wujiang District as the study unit and relying on the ArcGIS 10.4 software platform, the spatial autocorrelation analysis method was used to quantitatively analyze and study the spatial differentiation of the regional RILP. The I index was highly significant throughout the spatial autocorrelation analysis of RILP in the study area in 2020. Moran's I index was 0.110, indicating a significant positive spatial correlation at a 99.9% confidence level. There is apparent spatial clustering of high and low values across the whole region space, as evidenced by the fact that rural industrial land with a higher RILP is adjacent to higher rural industrial land, and lower rural industrial land tends to be adjacent to lower rural industrial land.

To more intuitively observe the spatial differentiation of the RILP, we visualized the different clustering patterns of high and low values of RILP in the study area by drawing LISA clustering maps. The high–high cluster is mainly concentrated in the villages in the north of Lili Town, the middle of Pingwang Town, the northeast direction of Shengze Town, the southwest direction of Zhenze Town, and around the centralized construction area in Qidu Town. The low–low cluster is heavily concentrated in the villages in the centralized construction area of Hengshan Street and the west side of Lili Town, and the peripheral villages of Zhenze Town, especially in the marginal village zones of the township. The overall character of the area is high in the south and low in the north. In addition, many

high–low and low–high outliers exist in each township, resulting in inevitable heterogeneity in the overall spatial distribution (Figure 7).

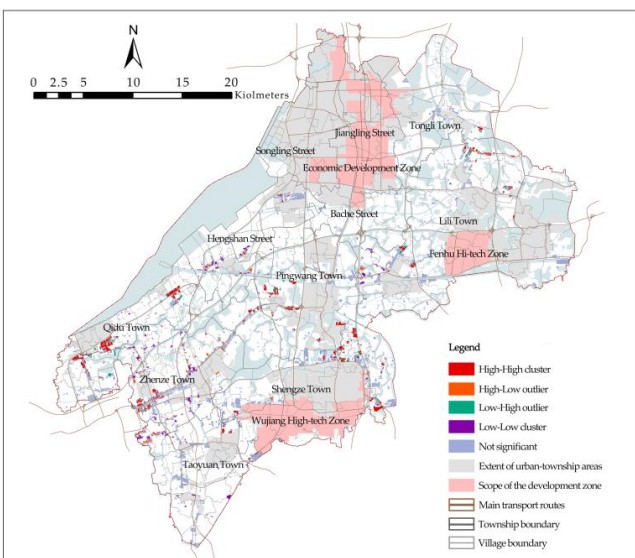

**Figure 7.** LISA agglomeration map of RILP.

We adopted the local spatial autocorrelation method to further reveal the cold and hot spots of RILP in the study area. This was combined with the Getis–Ord $G_i$* index value of ArcGIS 10.4 to measure the RILP and classify values into seven levels according to Jenks to generate a distribution diagram of cold and hot spots of RILP in Wujiang District (Figure 8). Overall, the Getis–Ord $G_i$* index hot spots (high-value areas) and cold spots (low-value areas) of RILP in Wujiang District are clustered. Specifically, the hot spots (high-value areas) of RILP are mainly located in the villages north of Lili Town, Pingwang Town, Shengze Town, and around the centralized construction area of Qidu Town. These areas have higher economic development levels, good location and transportation conditions, sufficient capital and labor, and more space for development. They have formed RILP hot-spot areas. The cold spots (low-value areas) are mainly distributed in the villages southwest of Hengshan Street, northeast of Zhenze Town, east of Lili Town, and northwest of Taoyuan Town. Due to economic, locational, traffic, and industrial base constraints, these areas form cold-spot areas of RILP.

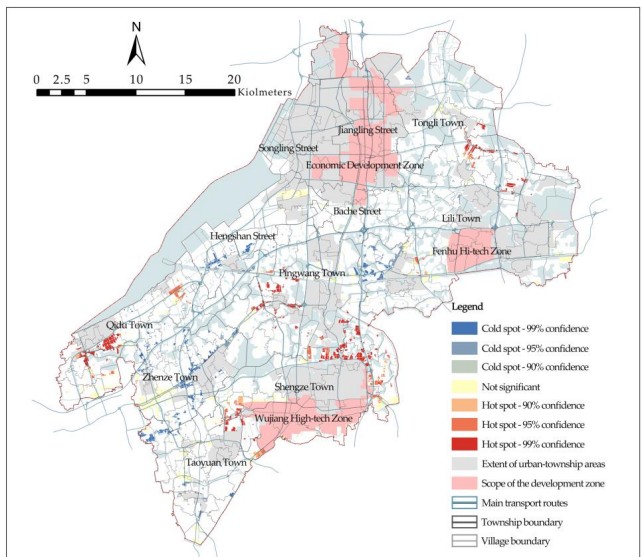

**Figure 8.** Distribution of cold and hot spots for RILP.

*4.3. Industry Evaluation of RILP*

At present, Wujiang District has formed four leading industries (silk textiles, equipment manufacturing, electronic information, and photoelectric communication), but the rural industry type is still dominated by traditional labor-intensive industries such as silk textile and equipment manufacturing. According to the Industrial Classification for National Economic Activities (GB/T 4754—2017), we summarized and compared the average performance evaluation values of 30 manufacturing industries on rural industrial land in Wujiang District. There are significant characteristics of polarization and differentiation (Figure 9). Technology-intensive industries are represented by electrical machinery and equipment manufacturing, automobile manufacturing, and general equipment manufacturing, and emerging industries are represented by pharmaceutical manufacturing and chemical fiber manufacturing, due to the solid technological innovation capability and high R&D investment of the enterprises themselves, resulting in the outstanding high value of the performance evaluation of the rural industrial land to which these industries belong. The labor-intensive industries represented by the comprehensive utilization of waste resources, agri-food processing, metal products, machinery, equipment repair, and the wine, beverage, refined tea, and food manufacturing industries ranked relatively low in performance due to the low scale, low efficiency, and low technology of the enterprises. Therefore, in the future, inefficient industrial land scattered in townships should be concentrated in various industrial agglomeration areas based on industry types to achieve industrial clusters and spatial layout optimization.

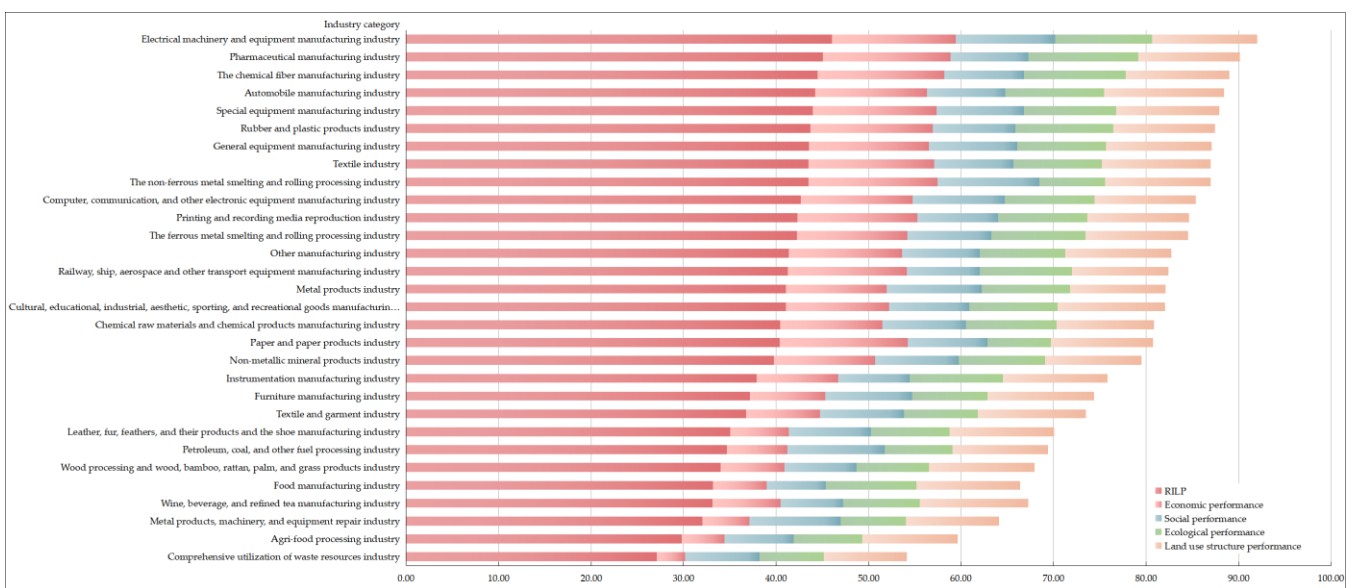

**Figure 9.** Statistical chart of RILP industry scores.

**5. Discussion**

*5.1. Policy Implications for Different RILP Types*

Based on the different disposal methods for industrial land, we combined the results of performance evaluation and spatial transformation policies and classified the evaluated plots into four categories: retained development, upgrading and transformed, function change, and remediation and withdrawal (Figure 10). We then linked them to the financial support policies for enterprises to implement differentiated control of the lands.

(1) The area of retained development land is approximately 1923.47 hm$^2$. This should retain industrial land attributes and continuously upgrade production capacity, improve relevant supporting services, and strictly prohibit industrial enterprises from being converted to other uses. We will provide enterprises with policies to appropriately lower taxes and profits, reduce the price of energy use, and encourage technological

innovation. We will also classify and specify indicators and development intensity standards for rural industrial land.

(2)   Upgrading and transformation land covers an area of approximately 636.13 hm$^2$, mainly comprising industries in urgent need of upgrading and transformation, with a focus on promoting industrial upgrading, transformation, and expansion. This will increase rural industrial land prices, development intensity, and other policies. Enterprises should also be guided to carry out specific land reclamation and renovation plans, and improve land use efficiency or output efficiency through technical reform, capital increase, production expansion, and other means. Enterprises that do not conform to industrial development guidelines should be required to carry out upgrading and transformation, especially in respect of energy-saving and emission-reduction technologies.

(3)   The area of function change land is approximately 828.61 hm$^2$, with an emphasis on promoting functional adjustment of land, relocation, and storage of indicators [61], and the need to be cautious to avoid large-scale "second-to-third" conversions leading to the relocation or demise of a large number of rural industrial enterprises. In addition, it is necessary to raise the price of land and energy use and set emission targets from industrial, commercial, and environmental aspects to force enterprises to transform and develop, and to increase the costs of their operations in strict accordance with relevant laws and regulations.

(4)   The area of remediation and withdrawal land is approximately 1198.82 km$^2$, and its expansion should be strictly controlled in the near future. The relocation and exit of industrial functions should be achieved through the relocation of indicators and relocation and consolidation. We should research and formulate policies to encourage the relocation of enterprises and the elimination of financial and tax concessions and support, and attention needs to be paid to the distribution of interests of all parties in the relocation and consolidation processes.

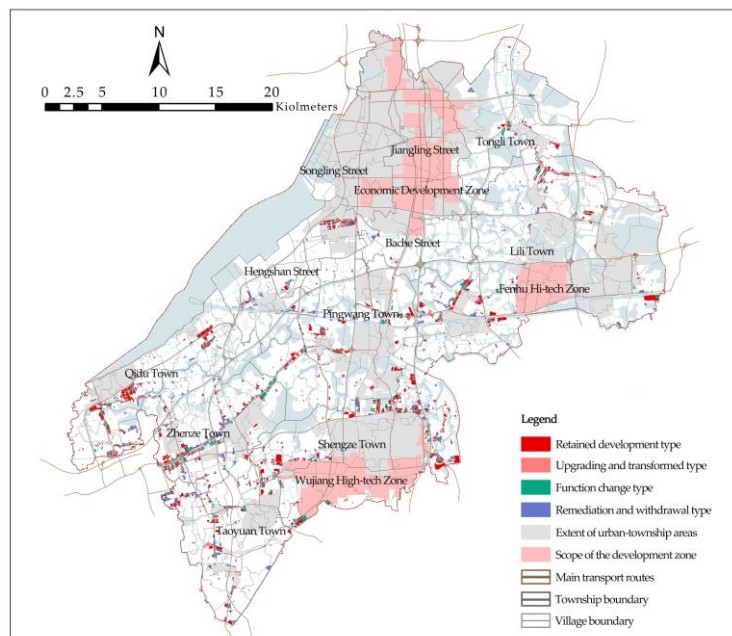

**Figure 10.** Spatial development classification of rural industrial land.

*5.2. Optimization Strategies for Rural Industrial Land Based on Performance Evaluation*

Based on the performance evaluation results, to identify the key problems of the rural industrial land use in the study area, we propose the following optimization strategies.

First, strategies should guide rural industrial spatial agglomeration. The spatially coordinated agglomeration of industrial land is a critical path by which to narrow the gap

between regions in terms of development quality and performance level, optimize the spatial layout of the territory, and promote the high-quality development of the manufacturing space [53]. Local governments in townships need to strengthen land use control at the macro level to achieve planning guidance and the spatial agglomeration of rural industrial land [61]. They can help industries with high-level clusters through the centralized construction of integrated, intensive industrial agglomeration areas. Therefore, it is suggested to set up 1–2 target-intensive industrial agglomeration areas in each township, and in principle, no new rural industrial land will be added. However, villages can use land indicators to promote industrial transfer and achieve the spatial agglomeration of rural industrial land at the township and higher levels.

Second, strategies can help the high-level clustering of rural industries. By optimizing and adjusting the rural industrial structure, we should give full play to the agglomeration effect of advantageous industries and transfer and eliminate low-value-added industries. (1) Through industrial base analysis and industrial chain assessment, we should identify the types of industries focused on encouraging, actively guiding, or phasing out, and support a differentiated positive and negative list of industries for the agglomeration areas. We should encourage and support the development of advanced manufacturing and emerging industries. (2) We should give full play to the agglomeration effect of advantageous industries, comprehensively upgrade the existing industrial technology level through equipment introduction and technology grafting, promote the high-end industrial chain, build a "community" of industrial chains, and exert their radiating effect. (3) We should transfer and eliminate low-value-added industries and phase out "low-capacity, high-energy-consumption" manufacturing industries such as paper and paper products; petroleum, coal, and other fuel processing industries; nonferrous metal smelting and rolling processing industries; metal products, machinery, and equipment repair industries. There should also be comprehensive utilization of waste resources.

### 5.3. Full Life-Cycle Supervision Mechanism Construction for Rural Industrial Land

Based on the unique cadastral code "BSM", and combined with the performance evaluation results, a dual-coupling platform of "industrial space platform + land trading platform" for different subjects was built using big data technology to explore the dynamic supervision and efficient management of the full life cycle of rural industrial land (Figure 11).

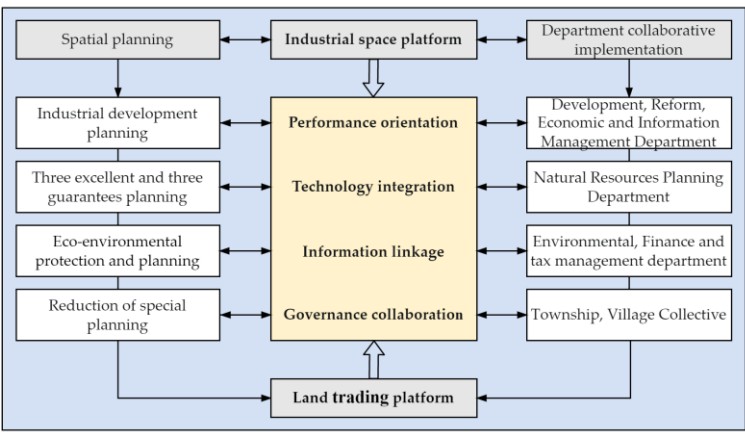

**Figure 11.** Construction of rural industrial land information linkage platform based on performance evaluation.

In the first stage, we established a database of industrial land projects based on the cadastral code that records information about rural industrial land, and relied on the industrial space platform to "link" the whole process of planning, approval, land supply, and registration [53] to realize the efficient linkage, sharing, and exchange of industrial information between management departments. By promoting the expression and coordination of spatial planning elements on the same drawing, the industrial spatial platform strengthens the integrated implementation of territorial planning and implements

"multiple compliance and reduction planning". In addition, the government will set up an online land trading platform for enterprises and the public, collect and release information on the demolition, transfer, and leasing of rural industrial land in the region, and provide market-oriented transfer services and share index transactions. The land trading platform takes into account the efficient linkage between the demand of industrial enterprises and the supply of land space, and realizes the accurate supply of rural industrial land. It is useful for correcting rural industrial land's inefficient sprawl and conserving the natural environment.

Based on a performance evaluation, the "full life-cycle regular physical examination" and dynamic assessment of rural industrial land and enterprises will be strengthened once every five years, and the evaluation results will be linked with cadastral codes and shared to the information linkage platform for a real-time update. Management departments can use the information linkage platform for all types of land online monitoring and real-time tracking.

## 6. Conclusions

Against the current background of controlling the expansion of construction land, activating stock, and "linking the increase in urban construction land with the decrease in rural construction land", we should strive to solve the problem of inefficient rural industrial land use to ensure the intensive and economical use of collective construction land [47,62,63]. We have proposed an analytical framework for intensive-use-oriented performance evaluation for rural industrial land based on smart shrinkage theory. This framework better fits the connotations of industrial transformation and intensive use development orientation within the context of stock and reduction planning. Based on this framework, we constructed a parcel-microscale RILP evaluation index system in four dimensions: economic performance, social performance, ecological performance, and land use structure performance. We expanded the research perspective on intensive rural industrial land use. Further, corresponding optimization strategies were proposed, which are useful for boosting rural industrial transformation and upgrading intensive rural industrial land use.

In this study, we selected Wujiang District as an empirical case. We used rural industrial land as the research unit and measured the performance of 4780 rural industrial land blocks in the study area. The main findings are as follows: (1) The performance evaluation scores were assigned to five grades according to Jenks: low, medium-low, medium, medium-high, and high performance. The grades covered areas of 1198.82, 828.61, 636.13, 999.28, and 924.19 hm$^2$ and accounted for 26.13%, 18.06%, 13.87%, 21.78%, and 20.15% of the total rural industrial land area in the study area, respectively. There are close to normal distribution characteristics and clear spatial differences in the RILP grades. (2) Moran's I index of RILP is significant, and the performance distribution shows high spatial agglomeration. Cold- and hot-spot detection indicates that the hot spots (high-value areas) and cold spots (low-value areas) of RILP are clustered. (3) The average performance evaluation values of 30 manufacturing industries on rural industrial land in the study area have significant characteristics of polarization and differentiation. The results not only reflect the RILP microscale level, but also better reflect the differences in rural industrial land in space and industry.

The performance evaluation index selection of rural industrial land is influenced by the socioeconomic development in rural areas. Compared with previous studies, the index selection in this study was more comprehensive and focused on the selection of spatial element indices, which can provide a reference for future scholars to conduct similar studies. Further, the performance evaluation method of rural industrial land established in this study can be further improved into a two-step evaluation method of "screening first and measuring later". Before the formal evaluation, it is suggested to select rural industrial land blocks that are not suitable for unified evaluation, so as to improve the practical application value. This evaluation method can also be used to simulate industrial

land reduction schemes in rural land consolidation planning and design [60]. Additionally, performance evaluation can help grasp the background of land resources and provide basic support for the preliminary analysis, implementation assessment, and dynamic monitoring of territorial space planning [64].

We constructed an RILP evaluation index system based on smart shrinkage theory, which is innovative. On the one hand, the study enriches the assessment methodology for the identification and withdrawal of inefficient rural industrial land against the background of rapid urbanization, and provides tools for further optimizing the use of industrial land in regions with developed township and village enterprises, such as the Yangtze River Delta and the Pearl River Delta. On the other hand, a performance judgment method that can be compared with similar types of districts or townships will be established to improve planning coordination. However, there are still certain deficiencies. Firstly, due to the limitations in data acquisition, the evaluation index system in this study cannot cover the indices of governance structure, the willingness of enterprises to relocate, the willingness of villagers, planning control, and other subjects' dynamic performance. Moreover, since this performance evaluation method's construction was based only on the Wujiang District, its applicability may need to be further verified in subsequent studies.

Although rural industrial land use is fragmented and random in Wujiang, the performance evaluation at least helped encourage industrial land pooling at the village level. In the future, based on obtaining more data, the performance evaluation of rural industrial land can be combined with multiple evaluation time points to carry out dynamic performance evaluation. More empirical studies are required to explore the practicability of this performance evaluation index system and methodology in the same type of district/county, and future research should be extended to the Yangtze River Delta, Pearl River Delta, and other regions with developed TVEs.

**Author Contributions:** Conceptualization, all authors; methodology, X.Y.; software, X.Y.; validation, L.F. and C.L.; formal analysis, X.Y.; writing—original draft preparation, X.Y.; writing—review and editing, L.F. and C.L.; visualization, X.Y.; funding acquisition, L.F. and C.L. All authors have read and agreed to the published version of the manuscript.

**Funding:** This research was funded by the National Natural Science Foundation of China (Nos. 52078316, 51978432) and the Center for Chinese Urbanization Studies, Soochow University and the Collaborative Innovation Center for New Urbanization and Social Governance (Nos. NH33712522).

**Institutional Review Board Statement:** Not applicable.

**Informed Consent Statement:** Not applicable.

**Data Availability Statement:** All data and models generated or used in the research process of this paper are presented and explained in the body of the article. The data or work not belonging to this paper have been appropriately cited or quoted as references.

**Acknowledgments:** The authors gratefully acknowledge the support of the funding.

**Conflicts of Interest:** The authors declare no conflict of interest.

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
