# Peer review of "Intensive-Use-Oriented Performance Evaluation and Optimization of Rural Industrial Land: A Case Study of Wujiang District, China"

_sustainability, doi:10.3390/su15118523_

Round 1

Reviewer 1 Report

Using the smart shrinkage theory, this manuscript constructing a new an analytical framework for intensive-use performance evaluation of rural industrial land to measure the performance of rural industrial land and proposes optimizing strategies for different rural industrial land types, which has certain reference value for improving the spatial performance of rural industrial land and promote industrial upgrading and land use transformation . However, the following problems still exist in the abstract, literature review, results and analysis and other aspects of the paper, and it is suggested to modify:

Comment 1: Abstract part of the paper lacks the theoretical significance of the description of the research work, in the description of the research results of logic chaos, lack of order, the characteristics and innovation of the paper is not prominent, the expression is not systematic and logical, suggest the author to refine the abstract of the paper.

Comment 2: In the introduction, the description of practical problems is relatively broad and does not focus on the theme of the paper, "Intensive-Use-Uriented Performance Evaluation and Optimization of Rural Industrial Land". There is insufficient research on the practical background of the research, and the practical problems targeted by the research are not clear, which leads to the lack of profound practical significance of the research in this paper. It is suggested to supplement.

Comment 3: The literature review in the introduction is not well organized and the topic is not focused enough. It is suggested to carry out literature review by referring to the logic of high-level papers, pay attention to the cohesion of the structure before and after and the refinement of language expression. The author is suggested to cite the following literatures:

[1]    Spatial and temporal patterns of urban land use efficiency in the Yangtze River Economic Belt from 2005 to 2014. [J] Journal of Geographical Sciences, 2018, 73(07): 1242-1252.

[2]    Land use dynamics driven by rural industrialization and land finance in the pen-urban areas of China: "The examples of Jiangyin and Shunde". [J] LAND USE POLICY, 2015, 45: 117-127.

Comment 4: In method part, the presentation of statistical data is too redundant . Attention should be paid to the standardization and refinement of language expression. It is suggested to refer to the expression of published high-level papers for language organization and expression form.

Comment 5: In terms of paper results, the authors attempt to break down and interpret the results from different angles, but ignore the overall logical coherence and consistency before and after. The interpretation of the research results is divorced from the results themselves, and does not start from the model results of empirical research, and combines theory and practice. At the same time, the purpose of different research results is not clear, resulting in the overall result system is more chaotic, and it is recommended to modify it.

Comment 6: It is suggested to further sublimate the discussion section, focusing on the analysis of the contribution and shortcomings of the article, appropriately proposing targeted suggestions, and future research directions.

Comment 7: The overall writing logic of the article needs to be strengthened, and the language expression needs to be further condensed and improved, paying attention to the coherence between the contexts. At the same time, pay attention to details, such as expression specifications, neat formatting and picture beautification.

none

Reviewer 2 Report

This is a manuscript dealing with an important subject. This is because there are rural industrial lands all over the world. This is because suggestions for the process of reforming the dual structure of urban and rural land use will continue to be essential in each country.

1. Smart shrinking theory made me worried that improving intensive land use would mean using more land for industrial and agricultural areas. However, according to the manuscript, it is written that the inefficient sprawl will be corrected and the accessible natural environment will be preserved. Should I understand this recommendation as a presentation of ideal optimization strategies? I would like you to show the conclusion part a little thicker.

2. Regarding land use, is there no "basic plan (Zoning, master plan or grand design) for setting land use areas" in the first place? That is the basis, and on top of that there is a reality (current state: de facto), and should we consider it a manuscript to improve that reality?

If that is the case, I would like you to touch on the "basic plan (Zoning, master plan and grand design) for setting land use areas". Are the authors not considering changes to this master plan?

3. About Land Trading Platform from Line520:

The idea of Full life cycle of rural industrial land is interesting. This is an idea that focuses on the role that the land should play, and is useful for correcting inefficient sprawl and conserving the natural environment. This part is written too succinctly, but I think many people will benefit from a little more detail.This is a manuscript dealing with an important subject.

Reviewer 3 Report

The paper is relatively comprehensive in its literature review, though, more literature review on Smart Shrinking theory should be added. In addition, for the "Methodology" section, I would like to see a few references for 3.3.1. Currently, there is only one. If no more, then there should be more explanation why the specific methods were chosen for filling what methodological gaps.   

On Page 8 Figure 4 shows important information on the spatial distribution patterns of existing rural industrial land, but other than the absolute land plots and %, no texts about the spatial distribution patterns. 

On Page 12 Table 4 and Figure 6: It is good to see all Table 4 mentioned are labeled on Figure 6. "Towns" are easy to see on the map but "Streets" are not. Although the streets are also labeled, no way to see where they are on the map. I suggest using Inset Maps to show them so more clear communications can be made. 

On Page 15 "5 Discussion": 5.1 section is weak as it does not provide what indicators were in the previous studies and what new indicators were chosen and why these new indicators were chosen. 

On Page 16 Figure 10: What is the "Scope of the development zone"? 

Round 2

Reviewer 1 Report

none

none